# Unique Interplay between Molecular miR-181b/d Biomarkers and Health Related Quality of Life Score in the Predictive Glioma Models

**DOI:** 10.3390/ijms21207450

**Published:** 2020-10-09

**Authors:** Rytis Stakaitis, Aiste Pranckeviciene, Giedrius Steponaitis, Arimantas Tamasauskas, Adomas Bunevicius, Paulina Vaitkiene

**Affiliations:** 1Laboratory of Molecular Neurooncology, Neuroscience Institute, Lithuanian University of Health Sciences, Eiveniu str. 4, LT-50161 Kaunas, Lithuania; giedrius.steponaitis@lsmuni.lt (G.S.); arimantas.tamasauskas@kaunoklinikos.lt (A.T.); 2Laboratory of Behavioral Medicine, Neuroscience Institute, Lithuanian University of Health Sciences, Eiveniu str. 4, LT-50161 Kaunas, Lithuania; aiste.pranckeviciene@lsmuni.lt (A.P.); adomas.bunevicius@lsmuni.lt (A.B.); 3Laboratory of Molecular Neurobiology, Neuroscience Institute, Lithuanian University of Health Sciences, Eiveniu str. 4, LT-50161 Kaunas, Lithuania; paulina.vaitkiene@lsmuni.lt

**Keywords:** glioblastoma, miR-181, prognosis, exosomes, quality of life

## Abstract

In the last decade, an increasing amount of research has been conducted analyzing microRNA expression changes in glioma tissue and its expressed exosomes, but there is still sparse information on microRNAs or other biomarkers and their association with patients’ functional/psychological outcomes. In this study, we performed a combinational analysis measuring *miR-181b* and *miR-181d* expression levels by quantitative polymerase chain reaction (qPCR), evaluating isocitrate dehydrogenase 1 (*IDH1*) single nucleotide polymorphism (SNP), and O-6-methylguanine methyltransferase (*MGMT*) promoter methylation status in 92 post-surgical glioma samples and 64 serum exosomes, including patients’ quality of life evaluation applying European Organization for Research and Treatment of Cancer (EORTC) questionnaire for cancer patients (QLQ-30), EORTC the Brain Cancer-Specific Quality of Life Questionnaire (QLQ-BN20), and the Karnofsky performance status (KPS). The tumoral expression of miR-181b was lower in grade III and glioblastoma, compared to grade II glioma patients (*p* < 0.05). Additionally, for the first time, we demonstrated the association between miR-181 expression levels and patients’ quality of life. A positive correlation was observed between tumoral miR-181d levels and glioma patients’ functional parameters (*p* < 0.05), whereas increased exosomal miR-181b levels indicated a worse functional outcome (*p* < 0.05). Moreover, elevated miR-181b exosomal expression can indicate a significantly shorter post-surgical survival time for glioblastoma multiforme (GBM) patients. In addition, both tumoral and exosomal miR-181 expression levels were related to patients’ functioning and tumor-related symptoms. Our study adds to previous findings by demonstrating the unique interplay between molecular miR-181b/d biomarkers and health related quality of life (HRQOL) score as both variables remained significant in the predictive glioma models.

## 1. Introduction

In the early stages of glioblastoma formation, usually no specific symptoms are present leading to its late detection, usually only when the tumor is already grown significantly and/or spread to other parts of the brain [1]. Glioblastoma multiforme (GBM) symptoms are not well defined and depend on the location of the tumor, but most commonly include headache, nausea, visual impairment, motor disorders, seizures, personality changes, or even disorientation, and very severe memory impairments in severe cases [2].

Standardized diagnosis of glioma consists of patient’s evaluation by computed tomography or magnetic resonance imaging, followed by histological analysis of the suspected tumor tissue [3]. However, even after histological examination, the characterized and grouped tumors often differ in their transcriptomal profile within the same malignancy group, which leads to complicated and limited-efficiency standardized treatment [4]. Therefore, it is necessary to determine the transcriptomal markers of gliomas in high sensitivity. Although vast amounts of genetic and epigenetic data from tumor tissue have been already collected and are publicly available from The Cancer Genome Atlas (TCGA) and other consortia, there is still a lack of epigenetic data from glioma patients’ serum exosomes, which could lead to improved glioma characterization and non-invasive diagnosis.

Intensive research over the last years proven the importance of miRNAs in the molecular biology of glioma [5]. Mature miRNAs are small non-coding RNA molecules which act as gene silencers in post-transcriptional manner [6]. Some of these small RNAs are associated with major depression, suicide behavior, and anxiety [7,8]. The small footprint of miRNAs and their ability to stimulate behavior changes makes miRNAs an attractive target analyzing varying psychological state of glioma patients.

The ability to detect circulating small RNAs in human blood has opened the vast potential for use of miRNAs as complication-free biomarkers for the diagnosis of various cancers [9,10]. However, subsequent studies have shown that most circulating miRNA are also highly expressed in different blood cells [11]. In order to avoid misleading results, the research strategy should only include miRNA analysis of circulating exosomes that are likely to be produced by highly invasive tumors such as glioblastoma and only those with very low leukocyte expression [12].

According to the bioinformatic analysis of Z. Yeng et al., the *miR-181* family is calculated to have more than 500 reliable targets. These targets are responsible for various biological processes such as cell proliferation, division, growth, and intercellular communication [13]. *miR-181* importance in oncology is observed in different types of cancer. A decrease in *miR-181a* expression was reported to cause downregulation of matrix metalloproteinase-1 and vascular endothelial growth factor expression in chondrosarcoma leading to a decrease in the tumor malignancy [14]. In glioblastoma cell lines and nude mice models, *miR-181b* has been shown to have onco-suppressor abilities, and its high expression is associated with a better outcome [15]. Downregulation of *miR-181a*, *miR-181b*, and *miR-181c* was observed in a small cohort of primary glioblastoma tissue, also suggesting miR-181 family involvement in glioblastoma development [16]. To further investigate the importance of different miR-181 family members in glioma, we decided to analyze *miR-181b* and *miR-181d* expression levels in a bigger patient cohort of glioma tissue and exosome samples.

Since glioblastoma still remains an uncurable disease, it is crucial to improve the quality of life of GBM patients and predict their quality of functioning after the surgery [17]. Therefore, this study included known and potential biomarkers of glioma, trying to better understand the molecular profile of gliomas and assess other microRNA connections with patients’ quality of life measurements. Furthermore, in this study, we tried to find a relationship between patients’ quality of life scores and exosomal *miR-181* levels. The assumption was that the radiologically non-detectable damage occurs in the healthy brain tissue surrounding the tumor. We hypothesized that these small, early damages to the surrounding tissue might evoke the variety of functional/psychological symptoms and could be caused by tumoral microRNAs transported in exosomes. To date, there was no information on *miR-181* tumoral or exosomal expression levels’ association with patients’ functional/psychological parameters.

## 2. Results

### 2.1. miR-181 Expression within Different Grades of Glioma

We evaluated both tumor and exosomal *miR-181b* and *miR-181d* (*miR-181b/d*) expression as a diagnostic biomarker for the identification of different grade gliomas. *MiR-181b* and *miR-181d* showed a tendency to be downregulated in grade 3 glioma in both tumor and exosome samples. However, a significant difference between grade 2 and grade 3 glioma was only detected when measuring *miR-181b* expression in tumor samples (*p* < 0.05) (Figure 1A). A vast distribution of *miR-181b/d* was detected in both GBM tumor and exosome samples, as was expected due to the heterogenic nature of grade 4 glioma.

Next, we wanted to compare *miR-181b/d* levels together with other known glioma biomarkers. Higher tumoral expression of both *miR-181b* (*p* < 0.05) and *miR-181d* (*p* < 0.01) was associated with isocitrate dehydrogenase 1 (*IDH1*) mutation (Figure 2A,B). *MiR-181d* level was significantly higher in GBM tumor tissues of patients with a *IDH1* R132H variant, which is primarily found in secondary GBMs, compared to *miR-181d* levels of GBM patients with a *IDH1* wild type (*p* < 0.001) (Figure 2C). Additionally, GBM patients that survived longer than an expected 12-month period and had an unmethylated O-6-methylguanine methyltransferase (*MGMT*) promoter also had a significantly higher *miR-181d* expression in their tumor tissue (*p* < 0.05) (Figure 2D). No significant results were observed while comparing exosomal *miR-181b/d* levels with different *IDH1* or *MGMT* patient groups.

### 2.2. miR-181 Expression and Functional Status of Patients

To investigate relationships between *miR-181b/d* levels, the functional status and symptom profile of patients’ correlation analysis was performed.

As can be seen in Table 1 and Table 2, *miR-181b/d* was significantly related to subjectively evaluated patients’ functioning; however, different trends were observed for tumoral and exosomal *miR-181* expressions. Tumoral *miR-181b* expression was positively correlated with better physical role and social functioning, as well as better general quality of life. Similarly, though non-significant trends were observed in the subsample of glioblastoma only, tumoral *miR-181d* expression was also positively related to physical functioning in total and glioblastoma samples. Exosomal *miR-181b* was not significantly related to any of the functioning indicators. Exosomal *miR-181d* showed a significant inverse correlation with physical and emotional functioning in the total sample; a similar but non-significant trend was observed in the glioblastomas subsample.

Several significant correlations were observed when analyzing relationships between *miR-181b/d* and patients’ reported tumor related symptoms. Higher tumoral *miR-181b* level was related to less expressed drowsiness in glioblastoma patients (Spearman rho = −0.30, *p* < 0.05). Tumoral *miR-181d* was related to greater seizure probability both in the total sample and in glioblastomas only (Spearman rho 0.26 and 0.29, respectively, *p* < 0.05). Exosomal *miR-181b* correlated positively with greater tumor related visual difficulties both in the total sample and glioblastoma patients only (Spearman rho 0.32, and 0.27, respectively, *p* < 0.05). Exosomal *miR-181d* also was positively related to vision impairment in the total and glioblastoma patient samples (Spearman rho 0.34 and 0.32, *p* < 0.04), with more expressed drowsiness in the total sample, and with a similar non-significant trend in glioblastoma patients (Spearman rho = 0.35, *p* < 0.05, and 0.28). Exosomal *miR-181* was negatively correlated with seizure probability, both in the total sample and glioblastomas only (Spearman rho −0.36 and −0.32, respectively, *p* < 0.05).

### 2.3. miR-181 Expression and Patients’ Survival outcome

#### 2.3.1. miR-181 Levels in Post-Surgical Glioma Tissue

We also analyzed *miR-181b* and *miR-181d* expression in glioma tissue and evaluated its effect on patients’ survival time after the tumor dissection. The survival analysis revealed no significant survival time differences between high and low *miR-181b* or *miR-181d* levels in all stages of glioma. Furthermore, the analysis was supplemented with patients’ *IDH1* and *MGMT* status; however, tumoral *miR-181* levels, within the same *IDH1* status patient group, also did not indicate any changes in patients’ survival. However, a strong tendency was observed comparing *miR-181d* tumor expression between GBM patients with methylated *MGMT* promoter (*p* = 0.065). Patients within this group had a 6.22 month longer median survival when tumoral *miR-181d* expression was higher than the cohorts median tumoral expression.

#### 2.3.2. miR-181 Levels in Glioblastoma Patients’ Serum Exosomes

Despite the fact that there was no significant difference between *miR-181* tumoral expression and glioma patients’ survival time, we wanted to see if the same result was reflected measuring *miR-181* in glioma patients’ serum exosomes. Interestingly, survival analysis revealed a difference between different *miR-181* expression groups. A noticeable difference was only detected in GBM patients when the cohort was grouped into low (lower than median) and high (higher than median) exosomal *miR-181* expression groups. GBM patients who had low *miR-181b* serum exosomal expression survived significantly longer compared to patients with a high exosomal expression (*p* = 0.017; *df* = 1; *χ2* = 5.629) (Figure 3A). Patient groups with different *miR-181d* or *miR-181b/d* expression showed only a tendency, indicating better prognosis for patients with low *miR-181d* (*p* = 0.239) or *miR-181b/d* (*p* = 0.08) expression in serum exosomes.

In addition, well known glioma biomarker *IDH1* mutation status was included in the survival analysis. However, analysis of *miR-181* expression in GBM patients with *IDH1* wild type did not reveal a more sensitive survival prediction (*p* = 0.049; *df* = 1; *χ2* = 3.871) (Figure 3B). No significant increase in the survival prediction was observed including *IDH1* or *MGMT* status comparing exosomal *miR-181d* expression levels.

Furthermore, we investigated the effect of *miR-181* exosomal expression differences among GBM patient ages. Samples of GBM patients were divided into two groups: younger than median cohort age (younger) and older than median cohort age (older) GBM patients. Older patients who had lower *miR-181b/d* exosomal expression showed a strong tendency (*p* = 0.086) surviving longer, compared to older patients with higher *miR-181b/d* exosomal expression—median survival of 15.6 and 7.65 months, respectively. The younger GBM patient group with higher *miR-181b/d* exosomal expression had a 2.3 month shorter median survival than the older GBM patient group with lower *miR-181b/d* exosomal expression.

In order to evaluate *miR-181b/d* exosomal expression as a prognostic biomarker, GBM patients were grouped into two specific groups: group A—GBM patients who were older than 55.3 years, had *IDH1* wild-type genotype, a hypermethylated *MGMT* promoter, and higher than median *miR-181b/d* exosomal expression; group B—the same previous criteria but with a lower than median *miR-181b/d* exosomal expression. A significant difference in patient survival time was observed within these two small GBM patient groups. Patients from group A (*n* = 7), on average, survived 2.36 times shorter than patients from group B (*n* = 4) (*p* = 0.025; *df* = 1; *χ2* = 4.989) (Figure 4).

Finally, a combinational analysis was performed to estimate the importance of tumoral and exosomal *miR-181* levels, *IDH1* and *MGMT* status, tumor related symptoms, quality of life index, and functional patients’ status for the survival outcome prediction. Decision tree classification was applied to evaluate the impact of measured features as a complex for patient survival as well as to estimate its importance. The overall accuracy of the tree classifier was 82.2%. The short survival subgroup (< 16.85 months) prediction accuracy was higher (90.6%), while the long survival group (≥16.85 months) showed slightly poorer accuracy (67.6%). Although the long survivor subgroup sample size was smaller (37 vs. 64), the prediction accuracy was lower indicating that the subgroup exhibits greater heterogeneity, in terms of analyzed features, as compared to short survivors. The decision tree classifier predicted the highest possibility rate of short post-surgical survival time for the glioma patients with the combination of *IDH1* wild-type genotype, lower *miR-181d* tumoral expression, higher *miR-181b* tumoral expression, and weaker tumor related symptoms. The highest probability of longer survival was associated with the combination of *IDH1* mutation (R132H), severe tumoral symptoms, and higher *miR-181b* exosomal expression (Figure 5).

## 3. Discussion

The *MiR-181* family is strongly associated with glioma and glioblastoma development, according to other in vivo and in vitro studies [15,18]. Multiple interactions of various mRNA and lncRNAs with *miR-181* family members indicate the importance of this microRNA in glioma [19]. Additionally, *miR-181b* could be involved in the regulation of tumorigenesis and epithelial to mesenchymal transition of glioma [15]. Decreased expression of *miR-181b* has been shown to stimulate cell proliferation, migration, and invasion, in addition to its ability to regulate chemosensitivity of temozolomide [20,21]. In our study, we found that the expression of *miR-181b* differs among malignancies of glioma, thus indicating that *miR-181b* expression could be associated with the grade of glioma. According to our data, *miR-181b* tumoral expression is downregulated in higher grade gliomas, compared to lower grade, which is consistent with other studies [22,23].

*MiR-181d* is another miRNA that belongs to the *miR-181* family, and its low expression levels are related to poor patient survival, suggesting the important role of *miR-181d* and its potential as a prognostic factor for glioblastoma patients [24]. In particular, W. Zhang and colleagues showed that *miR-181d* targets *MGMT* and downregulates it, leading to better response to temozolomide treatment [18]. In addition to the findings of W. Zhang et al., our study showed a noticeable tendency of prolonged survival time in GBM patients with methylated *MGMT* promoter and higher *miR-181d* tumoral expression levels. These results strengthen the suggestion that *miR-181d* is activated during *MGMT* promoter methylation processes, in order to suppress *MGMT* oncogenic activity in GBM patients [25].

A wide spectrum of both *miR-181b* and *miR-181d* expression levels was observed in serum exosomes and tumor tissue samples of GBM patients. The main reason for this could be the extreme heterogeneity of the glioblastoma. Recent studies have suggested that tumoral *miR-181* expression could help indicate different subgroups of GBM [15,26]. Our results clearly show that *miR-181b* and *miR-181d* expression are not consistent in GBM tissue, but further analysis and standardized GBM subgroup evaluation guidelines have to be performed in order to apply these data for GBM subgrouping.

This study reveals, for the first time, the prognostic potential of measuring *miR-181b* and *miR-181d* expression in GBM patients’ serum exosomes. Interestingly, in our study, exosomal *miR-181b* expression showed completely different predictive association to tumoral *miR-181d* levels. Longer survival time was observed in GBM patients with a lower exosomal *miR-181b* expression. Usually, a similar expression pattern is detected in tumor tissue and serum exosomes [27,28]. A different prognostic association of *miR-181b* expression in GBM post-surgical tissue and serum exosomes could be explained by a prevention of *miR-181b* packaging to exosomes in the tumor cells and exporting them out of the tumor’s environment. One could consider that *miR-181b* expression is promoted in GBM cells as a defense mechanism against tumor development. At the same time, tumor cells would try to export this onco-suppressive *miR-181b* out of the GBM cells in order to survive. In that case, an indication of a good prognosis would be high tumoral and low exosomal expression of *miR-181b*, which we observe in our single and combinational analyses. However, this theory should be tested thoroughly, including functional analysis of *miR-181b*.

Molecular research traditionally relies on very formal outcome measures such as overall survival or progression-free survival. However, none of these variables reflect the current health status of a patient, the symptom burden, or the quality of his or her functioning. It is known that decreased health related quality of life (HRQOL) in glioma patients is a sensitive predictor of shorter survival [17,29]. Recent meta-analysis by Coomans et al. [30] demonstrated that some HRQOL variables were independent predictors of overall patient survival and progression-free survival in glioma patients. Significant correlation was reported between deterioration of HRQOL scores and tumor progression in glioblastoma patients in longitudinal studies [31]. Thus, HRQOL is a very informative outcome measure as it reflects the subjectively perceived burden of tumor-related symptoms at the moment of assessment, and also, it has predictive value for long-term overall survival prognosis [32].

However, patient-centered outcome measures are rarely investigated in relation to biomarkers. Only a few studies have tried to associate molecular biomarkers with patients’ quality of life measurements. The work of S. V. Chatzikyriakou et al. suggests that the levels of circulating collagen metabolites could be used as a quality of life indicator for chronic heart failure patients [33]. Similar studies were carried out by S. Kay et al., who revealed changes of matrix metalloproteinase levels in idiopathic pulmonary fibrosis patients with different HRQOL scores [34], and by J. Hu et al., whose work linked serum *miR-206* levels to the quality of life of Duchenne muscular dystrophy patients [35]. Besides our previous work on *miR-34a* [36], brain cancer patients’ quality of life and its association with circulating biomarkers have only been investigated by A. Bunevicius et al., whose study suggests the importance of free triiodothyronine and thyroid stimulating hormone levels [37]. All of this research indicates the possibility of patients’ quality of life prediction in various diseases and invites us to look at biomarkers from a patients’ psychological and functional point of view.

To the best of our knowledge, this study investigated relationships between *miR-181b* and *miR-181d* and glioma patients’ functioning for the first time. Functioning was assessed using HRQOL measures as well as by clinical evaluation performed by the treating neurosurgeon. Both tumoral and exosomal *miR-181b/d* expressions were weakly, but significantly related to patients’ reported functioning and symptoms. Tumoral *miR-181b/d* expression showed the tendency towards correlation with better functioning, while exosomal *miR-181d* were related to lower physical functioning and a slightly more negative tumor-related symptoms profile. Exosomal *miR-181d* was statistically significantly correlated with a smaller probability of epileptic seizure; still, this finding could be interpreted as a negative indicator since seizures are reported to be related to longer survival in glioma patients due to earlier tumor diagnosis and initiation of treatment [38]. In line with previous findings, HRQOL and subjectively reported tumor-related symptoms were significant independent predictors in the combinational analysis of survival outcome prediction. However, our study adds to previous findings by demonstrating the unique interplay between molecular *miR-181b/d* biomarkers and HRQOL, as both variables remained significant in the predictive models. These findings encourage further research on molecular markers and HRQOL connections.

Finally, the combinational analysis revealed the importance of both tumoral and serum *miR-181* transcript levels in predicting glioma patients’ post-surgical outcomes. The decision tree classifier revealed that *miR-181* played an important part in different predictive subgroups. In the scenario of an *IDH1* wild-type patient, both exosomal *miR-181* had no significant influence on patients’ outcome prediction; instead, the tumoral *miR-181* played an important part, especially the lower tumoral *miR-181d* level, which was the second most important factor for patients’ short survival prediction. Interestingly, only higher exosomal *miR-181b* levels, but neither levels of tumoral *miR-181*, were selected as a major factor predicting longer survival in the context of patients with *IDH1* mutation and more expressed tumoral symptoms. These findings indicate the possible interplay between *IDH1* and the regulation of tumoral/exosomal *miR-181* transcript levels and could serve as an additional factor for other radiological- and clinical-data-based prediction models [39,40].

It is important to mention that the study cohort was slightly younger and did not have the usual 1.57 to 1 (male to female) gender ratio. Additionally, due to difficult microRNA detection in low amounts of serum exosomes, some of the patients’ exosomal samples were unsuited for quantitative polymerase chain reaction (qPCR) analysis leading to a smaller data set. However, this study shows the importance of the *miR-181* family in GBM patients’ outcome, and it is one of the first studies evaluating the influence of exosomal *miR-181b* and *miR-181d* expression levels on GBM patients’ outcome and their quality of life prediction.

In conclusion, the findings of our study suggest that elevated *miR-181b* exosomal expression can indicate significantly shorter post-surgical survival time for GBM patients. Like other researchers, we demonstrate *miR-181b* and *miR-181d* expression decrease during glioma progression. More importantly, both tumoral and exosomal *miR-181* expression levels were related to patients’ functioning and tumor-related symptoms. Furthermore, glioma patients’ quality of life index, their tumor-related symptoms, *IDH1* status, and tumoral *miR-181b* levels are important factors predicting patients’ survival time. Furthermore, adding GBM patients’ *MGMT* promoter methylation, age, and exosomal *miR-181b* expression information improves predictive significance and should be considered in all future research regarding predictive exosomal biomarkers for glioblastoma patients.

## 4. Materials and Methods

### 4.1. Study Cohort

The patients’ age on the day of the surgery varied from 24.6 to 80.0 years with a median of 55.3 and an average of 54.9 years. The cohort consisted of 55.5% males and 45.5% females.

The study cohort reflected common glioma patients’ molecular and survival characteristic. Patients who had *IDH1* mutation had a 11.1-month longer median survival compared to patients with *IDH1* wild-type (*p* = 0.008; *df* = 1; *χ2* = 6.855). Patients who were younger than 55.3 years showed a 10.1-month increase in median survival compared to older patients (*p* = 0.002, *df* = 1, *χ2* = 10.077). The median survival of GBM patients was 12.3 months in contrast to 22.4 months for lower grade glioma patients (*p* < 0.001; *df* = 1; *χ2* = 19.9).

### 4.2. Samples

The research was performed in accordance with the Lithuanian regulations, principles of the Helsinki and Taipei Declarations. Written informed consent was obtained from every patient and protocols used in this work were evaluated and approved by the Ethics Committee of Kaunas region, Lithuania (protocol: L6.1-07/09, permission code: P2-9/2003, date: 10 October 2010; and protocol: BE-10-6, permission code: BE-2-3, date: 18 April 2016).

In total, 92 different grade glioma samples were surgically removed at Lithuanian University of Health Sciences Hospital Kaunas Clinics (LUHS KC) Neurosurgery department during the period of 2016–2019. The grade of glioma was histologically confirmed at LUHS KC Department of Pathological Anatomy: 15-stage II; 7-stage III; and 70-stage IV (glioblastoma/GBM). In addition, 64 matched blood serum samples were collected: 10–stage II; 7–stage III; and 46–GBM. Due to the rare occurrence of the disease, the maximum number of samples was included into the study.

### 4.3. Patients Functional Status Assessment

The functional status of patients was assessed prospectively before neurosurgery by a certified medical psychologist and neurosurgeon. Patients’ functional status was assessed using two different paradigms—asking patients subjectively to evaluate their health, symptoms, level of functioning, and general quality of life using standardized questionnaires; asking a neurosurgeon to evaluate the level of patient independent functioning by using a clinical scale. Three measures were used in the current study:

The European Organization for Research and Treatment of Cancer Quality of Life Questionnaire, EORTC QLQ-30 [41] is an internationally validated cancer-specific health related quality of life measure. The EORTC QLQ-C30 contains 30 items that were designed to assess global health status, functional status, role functioning, emotional functioning, cognitive functioning, social functioning, and various cancer related symptoms. In the current study, we used functional scales and the total score as indicators of subjective patients’ functioning [42,43].

The European Organization for Research and Treatment of Cancer Quality of Life Questionnaire, Brain cancer module, EORTC QLQ-BN20 [44] was used to evaluate subjectively reported brain tumor related symptoms. The QLQ-BN20 is a 20-item self-rating instrument. It addresses future uncertainty, visual disorder, motor dysfunction, communication deficits, and other common brain tumor-related symptoms.

The Karnofsky performance status scale (KPS) [45] was used for assessment of functional status. The KPS is an 11-point rating scale that is designed to measure a patient’s ability to carry out his/her normal activities and dependence on help and nursing care.

Data on patients’ functional status evaluated by the neurosurgeon were available for 77 patients (83.7%); psychological assessment was performed for 75 (81.5%) patients from a 92 brain tumor samples cohort, and for 52 (81.3%) patients from a 64 matched blood serum samples cohort.

### 4.4. DNA Isolation

DNA was extracted from ~40 mg frozen tumor tissue using the desalting method with chloroform, and Proteinase K. DNA concentration was measured with a NanoDrop 2000 system (Thermo Fisher Scientific, Cat. #: ND-2000, Wilmington, DE, USA).

### 4.5. IDH1 Mutation and MGMT Promoter Methylation Analysis

IDH1 gene mutation in gliomas-R132H was analyzed in all the specimens applying custom TaqMan SNP genotyping assays. PCR was carried out consisting of TaqMan Universal Master Mix II (Thermo Fisher Scientific, Cat. #: 4440047, Carlsbad, CA, USA), TaqMan probes, and 20 ng purified tumor DNA. All the procedures were accomplished according to the TaqMan chemistry manufacturer recommendations. Fluorescence was measured with a 7500 Fast Real-Time PCR system (Applied Biosystems, Cat. #: 4351107, Foster City, CA, USA).

MGMT promoter methylation status was determined using methylation-specific PCR (MSP). The reaction consisted of 7.5 µL Hot Start PCR Master Mix with Hot start Taq DNA polymerase (Thermo Fisher Scientific, Cat. #: EP0701); 4.5 µL nuclease-free water; 1 µL (10 pmol/µL) of each primer, specific to methylated/unmethylated promoter; and ~20 ng of bisulfite-treated DNA as a template. Primer sequences for the methylated MGMT sequence were 5′-GGACGTTAAGGGTTTAGAGC-3′ (sense) and 5′-CAATACACGACCTCGTCAC-3′ (antisense), and for unmethylated—5′-GGATGTTAAGGGTTTAGAGT-3′ (sense) and 5′-CAATACACAACCTCATCAC-3′ (antisense). Additionally, three controls were performed: positive—“Bisulfite converted Universal Methylated Human DNA Standard & Control primer” (Zymo Research, Cat. #: D5015, Irvine, CA, USA); negative—bisulphite treated human blood lymphocytes DNA; and water control (no template control). MSP was performed in 38 cycles with the following conditions: Taq Polymerase activation at 95 °C for 5 min, denaturation at 95 °C for 15 sec, annealing at 59 °C for 30 sec, extension at 72 °C for 15 sec, and final extension at 72 °C for 5 min. Products after amplification were visualized using agarose gel electrophoresis. Each sample methylation status was evaluated according to visible signals and documented using a 0 (unmethylated) and 1 (methylated) system.

### 4.6. RNA Isolation and cDNA Synthesis

In total, 20–40 mg of frozen, post-surgical tumor samples was mechanically grinded and homogenized with ultrasonication at 20% amplitude for 1 second on/off pulsation prior to enriched small RNA extraction using a mirVana miRNA Isolation Kit (Thermo Fisher Scientific, Cat. #: AM1560). An amount of 250–1250 µl of frozen serum samples was used for exosome extraction with exoEasy Maxi Kit (Qiagen, Cat. #: 76064, Valencia, CA, USA) including cel-miR-39-3p spike-in (0.0065 ng) in each sample after exosome collection step. The quality and quantity of extracted microRNAs were evaluated with a Small RNA analysis kit (Agilent, Cat. #: 5067-1548, Santa Clara, CA, USA) and NanoDrop 2000. In order to be able to analyze the broad range of micro RNAs, 10 ng extracted RNA was synthesized to cDNA with a TaqMan Advanced miRNA cDNA Synthesis Kit (Thermo Fisher Scientific, Cat. #: A28007, Pleasanton, CA, USA). Pre-amplified and 10 times diluted cDNA was used for micro RNA expression analysis afterwards.

### 4.7. Micro RNA Expression Analysis

QPCR reaction consisted of TaqMan Fast Advanced Master Mix (Thermo Fisher Scientific, Cat. #: 4444557, Austin, TX, USA), hsa-miR-181b-5p (Assay ID: 478583_mir) or hsa-miR-181d-5p (Assay ID: 479517_mir) probes, and 3 µl diluted cDNA. Gene expression was measured on a 7500 Fast Real-Time PCR system using a fast cycling program. In addition, 4 endogenous micro RNAs were used for data normalization: hsa-miR-191-5p (Assay ID: 477952_mir), hsa-miR-361-5p (Assay ID: 478056_mir), hsa-miR-345-5p (Assay ID: 478366_mir), and hsa-miR-103a-3p (Assay ID: 478253_mir). Additional levels of spike-in were measured in serum exosome samples (Assay ID: 478293_mir) (Thermo Fisher Scientific, Cat. #: A25576, Pleasanton, CA, USA).

For each sample, relative quantitation of hsa-miR-181b-5p and hsa-miR-181d-5p was calculated according to the Equation (1):(1)$$\Delta Ct_{target\;miR}=\;Ct_{target\;miR}-\;\sqrt[{4}]{Ct_{miR191}\times Ct_{miR361}\times Ct_{miR345}\times Ct_{miR103a}}$$

An additional normalization step was applied to serum exosome samples normalizing to spike-in cel-miR-39-3p levels according to the Equation (2):(2)$$\Delta Ct_{target\;miR}=\;\Delta Ct_{target\;miR}\times \;\left(\frac{\bar{Ct_{miR39}}}{Ct_{miR39}}\right)$$

### 4.8. Statistical Analysis

Kaplan–Meier estimation, using a log-rank test was performed to evaluate patient groups during survival analysis. A Student’s independent t-test was applied evaluating the difference between two groups, and One-Way ANOVA with Bonferoni correction was applied for the comparison of three or more groups. Spearman correlation was used to evaluate relationships between miRNA expressions and the functional status of the patients. Decision tree classification analysis was performed using the CRT algorithm with Gini method nonlinear combinations. The significance level was defined as *p* < 0.05 (*).

## Figures and Tables

**Figure 1 ijms-21-07450-f001:**
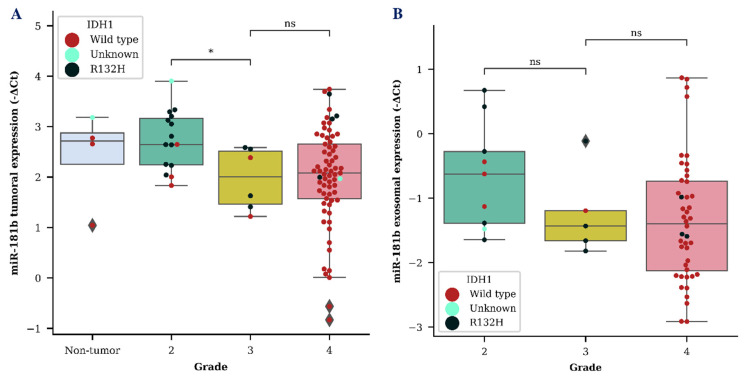
*MiR-181b* tumoral (**A**) and exosomal (**B**) expression levels within different grade gliomas. Colored dots represent different isocitrate dehydrogenase 1 (*IDH1*) C. 395G>A (R132H) variant status in the cells of the brain tissue. The box squares represent the data within 25 and 75 percentiles; the line in the middle shows the median.

**Figure 2 ijms-21-07450-f002:**
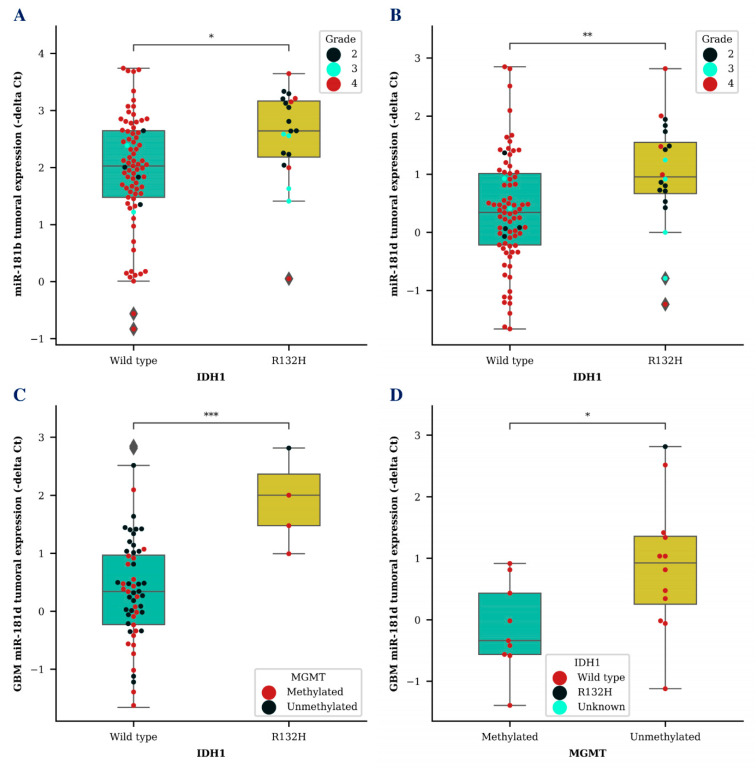
*MiR-181b* and *miR-181d* tumoral expression level association with known glioma biomarkers. Tumoral *miR-181b* (**A**) and *miR-181d*
**(B**) expression differences between glioma patients (**A**,**B**) or GBM patients (**C**) with *IDH1* wildtype and *IDH1* R132H variant. Tumoral miR-181d levels in GBM patients with methylated *MGMT* promoter and unmethylated *MGMT* promoter status (**D**). Colored dots represent different *IDH1* R132H variant (**D**) or *MGMT* promoter methylation status (**C**) and glioma grade (**A**,**B**). The box squares represent the data within 25 and 75 percentiles; the line in the middle shows the median.

**Figure 3 ijms-21-07450-f003:**
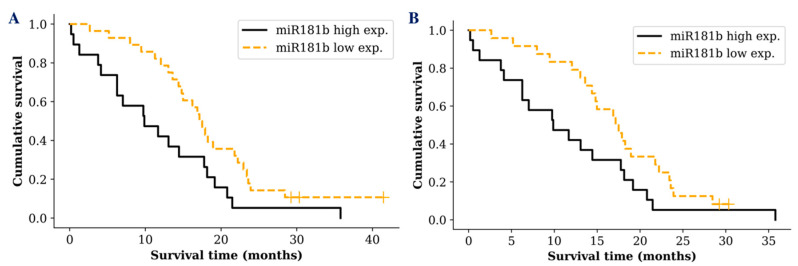
Kaplan-Meier survival curves comparing higher and lower *miR-181b* exosomal expression levels in: **A**—all GBM patients (*p* = 0.017); **B**—GBM patients with *IDH1* wild type (*p* = 0.049). Censored cases indicated by a vertical line.

**Figure 4 ijms-21-07450-f004:**
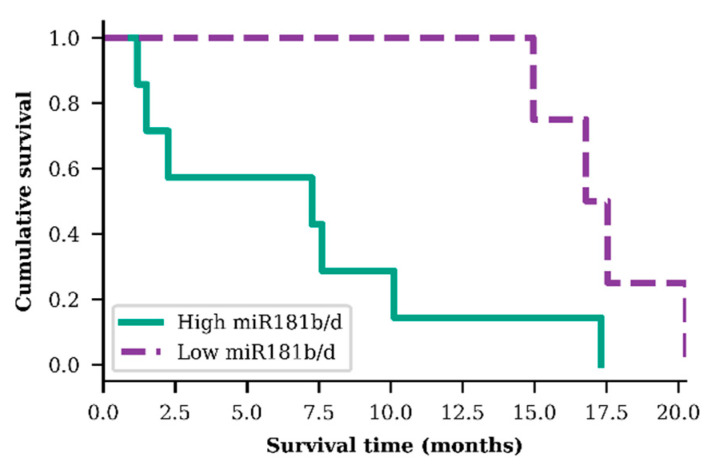
Survival analysis of both exosomal *miR-181b* and *miR-181d* high and low levels in older (>55.3 years) glioblastoma patients, diagnosed with *IDH1* wild type and methylated *MGMT* promoter (*p* = 0.025)

**Figure 5 ijms-21-07450-f005:**
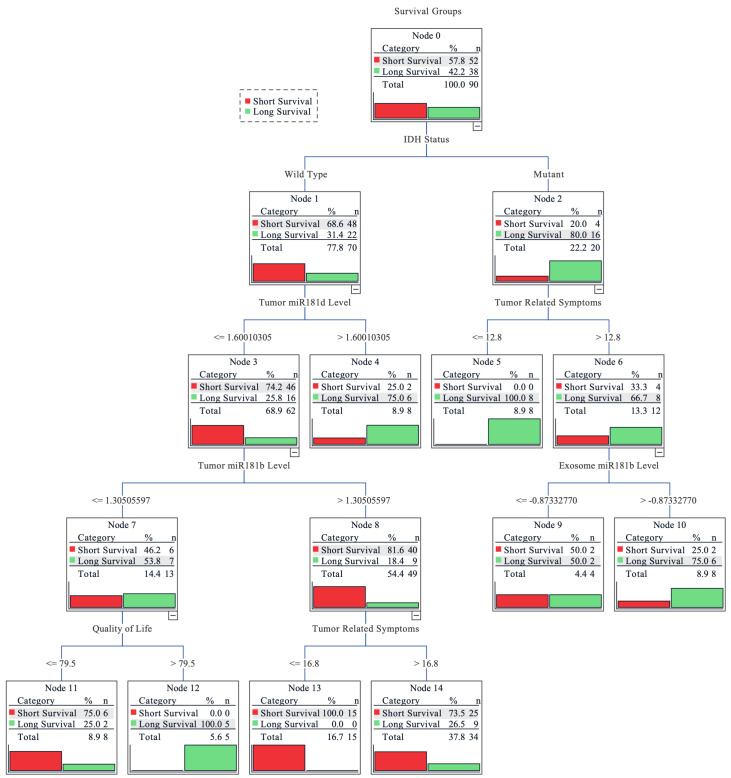
Decision tree for predicting grade II-IV glioma patients’ survival. Grouped into two subgroups according to the cohort survival mean: <16.85 months—short survival; ≥16.85 months—long survival. The earlier factor appearance (vertically going from top to bottom) shows its higher importance to the prediction model. Values on the lines indicate the factor value at which the algorithm divided the factor groups. For *miR-181* expression levels, the fold change value was used; the higher tumor related symbol score reflects more pronounced symptoms and the higher quality of life score indicates better functional and psychological well-being of the patient.

**Table 1 ijms-21-07450-t001:** Correlations between *miR181b* expression and subjectively reported quality of functioning.

Subjectively Reported Quality of Functioning Groups	*miR-181b*			
	Tumoral		Exosomal	
	GBM only	Total sample	GBM only	Total sample
Global health	−0.02	0.05	−0.09	−0.05
Physical functioning	0.23	0.27 *	−0.08	−0.09
Role functioning	0.18	0.23 *	0.09	0.08
Emotional functioning	0.24	0.15	−0.21	−0.18
Cognitive functioning	0.07	0.12	0.03	−0.05
Social functioning	0.32 *	0.33 **	−0.02	−0.04
Summary Quality of Life Score	0.19	0.28 *	−0.12	−0.08
Karnofsky Performance Scale	0.09	0.08	0.10	0.05

* *p* < 0.05; ** *p* < 0.01.

**Table 2 ijms-21-07450-t002:** Correlations between *miR181d* expression and subjectively reported quality of functioning.

Subjectively Reported Quality of Functioning Groups	*miR-181d*			
	Tumoral		Exosomal	
	GBM only	Total sample	GBM only	Total sample
Global health	−0.03	0.02	0.04	−0.02
Physical functioning	0.29 *	0.32 **	−0.20	−0.27 *
Role functioning	0.07	0.13	−0.03	−0.03
Emotional functioning	0.10	0.07	−0.27	−0.38 **
Cognitive functioning	−0.07	0.01	−0.16	−0.25
Social functioning	0.13	0.18	−0.05	−0.20
Summary Quality of Life Score	0.06	0.17	−0.15	−0.27
Karnofsky Performance Scale	−0.04	0.00	0.08	−0.04

* *p* < 0.05; ** *p* < 0.01.

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
