# Peer review of "Unique Interplay between Molecular miR-181b/d Biomarkers and Health Related Quality of Life Score in the Predictive Glioma Models"

_ijms, 2020, doi:10.3390/ijms21207450_

Round 1

Reviewer 1 Report

Overall I find this study highly correlative, it is not presented in the most attractive way. It is written quite carelessly, there are number of punctuation-type errors already in the Abstract (lines 19, 26 and 28).

From the major concerns:

  1. Why in Figure 1 only tumoral levels of miRNAs was shown. I would expecte that exosomal levels would be of more interest to the reader, they should be probably shown side by side.
  2. Table 1 is badly descibed "Relationships..." Aren't those just correlations, relationships is not very specific term.
  3. In many places a term "multivariate analysis" is repeated and to be honest this is the most important peace of analysis that I am missing. I would like to see s multivariate analysis testing the hypothesis that exosomal mir-181 expression is an independent prognostic factor and that measuring its level adds some clinical value to existing ones. Decision tree showed in Fig 5 is not testing this hypothesis.
  4. I have a strong doubts about Kaplan-Meier analysis from Fig3. How it is possible that both plots (A,B) have similar or even the same data points if one plot shows all GBM patients (A) and the other patients with IDH-mutant GBM (B). I assume, that there is an error in caption and you meant IDH-WT GBMs. Still it looks like nr of samples is almost identical on both graphs. Moreover on both plots lines cross, which violates lor-rank test conditions. I would advice to test some log-rank test alternative here as well. Moreover what are the thresholds for high/low expression and is the data censored/uncensored?
  5. Please revise the manuscript as there are plenty of errors - as written above, just in Abstract 3 punctuation errors spotted

Minor issues:

line 42 why "but only confirmed", please rephrase

line 42-44 - whole point of that sentence escapes my comprehension. IHX is useful tool to deal with gliomas, in this sentence it is kind of suggested that IHC is complicating things in diagnostic, whereas it is completely opposite in my opinion.

line 44-45 - I strongly disagree!! We have tons of genetic and epigenetic data for gliomas, from TCGA and other consortia!! We may just still struggle to make sense out of it

line 47 "whom" ?

line 54 - reference is missing

line 62 "influence to" ?

line 78 - Do you mean microRNAs from tumors that are released inside exosomes ? In current form it suggests that exosomes are producing miRs

2.1 Study cohort - it does not belong to Results section

line 103: IDH R132H is not a rare variant!!! It is rare in GBMs as it is found primarily in secondary GBMs. Please re-write!

line 140: As far as I can tell from the results you have presented, there is no "effect" of miR-181 on patients survival time. No proof that miR-181 is affecting patient's survival has been shown in this work

line 143: Sentence started with "But..."

In my opinion Discussion in the current form is to long and hardly digestable

line 210 Missing reference

Author Response

We have taken up the criticism and suggestions of the Reviewers. In the accompanying rebuttal letter are our point-by-point responses to each comment. In the revised manuscript, all changes have been indicated by the red-colored text.

Reviewer 2 Report

This manuscript entitled “Unique interplay between molecular miR-181b/d 2 biomarkers and health related quality of life score in 3 the predictive glioma models” shows that grade III and IV gliomas show lower expression of miR-181 and exosomal miR-181b/d expressions correlate worse overall outcome. This article is of value in that miR-181b/d is identified as novel functional factor in symptons and prognosis in glioma pateirnts. However, this reviewer has some concerns or question before recommendation for publication. These are listed below:

(1) Correlation (or no correlation) between tumoral and matched (serum) exosomal expressions of miR181b/d should be indicated.

(2) Higher grade gliomas show lower tumoral expression of miR181b, and tumoral miR-181b/d expressions correlate better functioning in several subjective parameters. On the other hand, exosomal miR-181b/d expressions correlate worse functioning in several subjective parameters (Table 1) and worse overall survival (Figure 3 and Figure 4). What is the biological significance of miR-181b/d? Whether are tumor suppressive or promoting function suggested?

(3) Only results of miR-181d are indicated in Figure 1 and Figure 3, and patients with higher or lower levels of both of miR-181b and miR-181d are analyzed in figure 4. It seems to be unfair.

Author Response

(The authors gave the same response as above.)

Round 2

Reviewer 1 Report

Dear Authors,

Thank you for making all corrections,

I have no further comments